# FLIm and Raman Spectroscopy for Investigating Biochemical Changes of Bovine Pericardium upon Genipin Cross-Linking

**DOI:** 10.3390/molecules25173857

**Published:** 2020-08-25

**Authors:** Tanveer Ahmed Shaik, Alba Alfonso-Garcia, Martin Richter, Florian Korinth, Christoph Krafft, Laura Marcu, Jürgen Popp

**Affiliations:** 1Leibniz Institute of Photonic Technology Jena e.V., Albert-Einstein-Str. 9, 07745 Jena, Germany; Shaik.TanveerAhmed@leibniz-ipht.de (T.A.S.); florian.korinth@leibniz-ipht.de (F.K.); christoph.krafft@leibniz-ipht.de (C.K.); 2Biomedical Engineering Department, University of California Davis, Davis, CA 95616, USA; aalfonso@ucdavis.edu; 3Institute of Physical Chemistry and Abbe Center of Photonics, Friedrich Schiller University Jena, Helmholtzweg 4, 07743 Jena, Germany; martin.richter@uni-jena.de

**Keywords:** genipin, cross-linking, FLIm, Raman spectroscopy, tissue engineering

## Abstract

Biomaterials used in tissue engineering and regenerative medicine applications benefit from longitudinal monitoring in a non-destructive manner. Label-free imaging based on fluorescence lifetime imaging (FLIm) and Raman spectroscopy were used to monitor the degree of genipin (GE) cross-linking of antigen-removed bovine pericardium (ARBP) at three incubation time points (0.5, 1.0, and 2.5 h). Fluorescence lifetime decreased and the emission spectrum redshifted compared to that of uncross-linked ARBP. The Raman signature of GE-ARBP was resonance-enhanced due to the GE cross-linker that generated new Raman bands at 1165, 1326, 1350, 1380, 1402, 1470, 1506, 1535, 1574, 1630, 1728, and 1741 cm^−1^. These were validated through density functional theory calculations as cross-linker-specific bands. A multivariate multiple regression model was developed to enhance the biochemical specificity of FLIm parameters fluorescence intensity ratio (R^2^ = 0.92) and lifetime (R^2^ = 0.94)) with Raman spectral results. FLIm and Raman spectroscopy detected biochemical changes occurring in the collagenous tissue during the cross-linking process that were characterized by the formation of a blue pigment which affected the tissue fluorescence and scattering properties. In conclusion, FLIm parameters and Raman spectroscopy were used to monitor the degree of cross-linking non-destructively.

## 1. Introduction

Animal-derived tissues are widely used in regenerative medicine surgeries and tissue engineering applications [1,2,3]. However, animal-derived tissues need to be decellularized and cross-linked before being implanted into human patients to reduce adverse antigen reactions and to increase resistance against enzymatic degradation [4]. In tissue engineering applications, longitudinal monitoring of tissues is a key factor to assess the quality of the grafts before implantation. Conventionally, collagenous tissue is evaluated by using labeling techniques like amino acid assays (hydroxyproline) to assess collagen content [5] and destructive techniques like high-performance liquid chromatography (HPLC) to assess the degree of cross-linking [6]. However, these techniques are not suitable for longitudinal monitoring in tissue engineering applications because the samples are dramatically compromised after just one measurement.

Optical imaging technology, on the other hand, offers the possibility to assess tissue content and properties in a tissue-conservative approach that enables longitudinal monitoring. Time-resolved fluorescence imaging has been used to study collagen and elastin content [7,8], collagen cross-linking [9], recellularization processes [10], and mechanical properties [9,11] of tissues in a label-free and non-destructive manner. Time-resolved fluorescence is a versatile imaging technique that can be used in a microscope (FLIM) and through flexible optical fibers (FLIm). The latter approach has fast imaging capabilities and enables imaging in complicated geometries (i.e., intravascular imaging). While the microscopic approach can be used to image deep into tissue when using multiphoton excitation [12], the fiber-based imaging technique is so far limited to UV excitation and constrained to surface imaging (low penetration depth ~100–200 μm). Additionally, UV light excites many of the tissue fluorophores, which have overlapping spectral emissions. This limits the biochemical specificity of FLIm. Validation of fluorescence lifetime information can be addressed with tissue staining approaches or exogenous labeling imaging. However, these are typically expensive and time-consuming, and the sample can no longer be used for regenerative medicine purposes. Combining FLIm with other label-free optical imaging modalities is a common strategy to increase the information extracted from tissue samples in a non-destructive manner. Raman spectroscopy, in particular, is well suited to enhance the biochemical specificity, as it provides a molecular fingerprint of tissues. Raman spectroscopy was used to detect cross-links in pericardium [13], dentin [14], bone [15], and collagen degradation [16]. The FLIm-Raman spectroscopy combination has already been used to study the biochemical properties of human atherosclerotic tissues [17] and collagen in bovine pericardium after glutaraldehyde cross-linking [18]. Furthermore, FLIm can be combined with other imaging modalities like OCT for added structural and depth information [19], second-harmonic generation/two-photon fluorescence (SHG/TPF) [20], or intravascular ultrasound (IVUS) [21] to investigate biochemical changes associated with the atherosclerotic process.

Tissue cross-linking is one of the processes of interest for longitudinal and non-destructive monitoring in tissue bioengineering. Typical cross-linkers include glutaraldehyde, formaldehyde, ethyl-(dimethyl aminopropyl) carbodiimide, and genipin [22]. Glutaraldehyde is commercially available and widely used in clinical applications despite its well-established cytotoxicity and susceptibility to calcification effects [23]. Alternatively, genipin (GE) is a water-soluble, natural cross-linker, 10,000 times less cytotoxic than glutaraldehyde, and it also improves the mechanical properties of tissue constructs [24,25,26,27]. GE is a colorless iridoid extracted from the gardenia fruit (*Gardenia jasminoides ELLIS*). It has previously been used to cross-link biological tissues, chitosan-based tissue equivalents, gelatin [28], bioadhesives [29], and amino acid detection agents [30].

GE cross-linking forms intra- and intermolecular cyclic cross-links within the collagen fibers of the extracellular matrix [25]. It reacts with the amine groups of lysine or hydroxy-lysine residues in collagen to form a nitrogen iridoid. After dehydration, an aromatic monomer is formed, which dimerizes to form GE cross-links. [27,31]. These GE cross-links absorb light in the range of 250 to 600 nm and fluoresce from 380 to 700 nm [32], thus GE cross-linked collagen turns blue and emits a redshifted fluorescence compared to collagen alone. Previous studies with Fourier transform infrared (FT-IR) and fluorescence spectroscopy associated the spectral changes to the GE cross-linker itself [33,34]

The goal of this study was to monitor the degree of GE cross-linking of decellularized and antigen-removed bovine pericardium (ARBP). To achieve this goal, we implemented a non-destructive approach based on FLIm and Raman spectroscopy to detect the biochemical changes that occur in the tissue during the cross-linking process. Lifetime and spectral maps rapidly acquired with the FLIm system reflected biochemical changes that were further specified by Raman spectroscopy, performed on selected locations of the tissue. Validation of the Raman spectral signature of the GE cross-linker was achieved with density functional theory (DFT). Finally, we established a relationship between FLIm parameters and Raman spectra to increase the overall biochemical specificity of this non-destructive and label-free approach. The ultimate goal of this methodology is to assess tissue bioengineering processes (i.e., cross-linking) while preserving tissue integrity for further biomedical use.

## 2. Results

### 2.1. Fluorescence Lifetime Imaging (FLIm)

#### 2.1.1. Lifetime Decrease upon GE Cross-Linking

The fluorescence lifetime of antigen-removed bovine pericardium (ARBP) decreased with cross-linking incubation time from 0.5 to 2.5 h. Figure 1a,c,e show the fluorescence lifetime maps of the GE cross-linked samples (GE-ARBP) in spectral band (SB) 1 (380–400 nm), SB2 (415–455 nm), and SB3 (465–553 nm), respectively. In SB1, the mean fluorescence lifetime decreases from 5.30 to 3.72 ns in 0.5 h, down to 2.91 ns in 1 h, and to 2.11 ns in 2.5 h (Figure 1b). A similar trend was observed in SB2 and the lifetime decreased from 5.38 to 3.12 ns in 0.5 h, down to 2.35 ns in 1 h, and to 1.60 ns in 2.5 h (Figure 1d). In SB3, the mean fluorescence lifetime abruptly dropped from around 5.04 to 2.17 ns in 0.5 h, and further decreased to 1.87 ns in 1 h, and 1.57 ns in 2.5 h (Figure 1f). A Mann–Whitney U-test indicated that the fluorescence lifetime differences between the GE-ARBP and untreated ARBP were statistically significant at every time point (*p* < 0.05).

#### 2.1.2. Spectral Redshift upon GE Cross-Linking

Intensity ratios were obtained dividing the fluorescence intensity of each SB to the summation of fluorescence intensities in all three measured SBs. Figure 2a–c show the fluorescence intensity ratio maps of the GE cross-linked samples in the three SBs. Intensity ratios are indicative of the spectral properties of the samples. Similar to fluorescence lifetimes, intensity ratios can monitor cross-linking at different time points. In SB1, the mean fluorescence intensity ratio decreased from 0.53 (untreated ARBP) to 0.24 a.u. in 0.5 h of incubation with GE, 0.16 a.u. at 1 h, and to 0.1 at 2.5 h. In SB2, the intensity ratio also decreased with increasing cross-linking incubation time, however, the differences with the untreated ARBP were unremarkable in all time points. In SB3, the mean fluorescence intensity ratio increased with increasing GE incubation time. Untreated ARBP had a mean intensity ratio of 0.08 a.u. and it increased to 0.41 a.u. at 0.5 h, to 0.51 a.u. at 1 h, and to 0.57 a.u. at 2.5 h GE cross-linking. Overall, a redshift was observed upon GE cross-linking (Figure 2d). This spectral shift progressed over the incubation time, being more pronounced on the last tested time point. The emission spectra of ARBP and GE cross-linked (at 2.5 h) confirmed the redshift (Figure 2e). Upon 355 nm excitation, untreated ARBP exhibited a maximum fluorescence emission at 390 nm, whereas the peak for GE-ARBP (2.5 h) was at 440 nm.

### 2.2. Raman Spectral Signature of GE Cross-Linked Tissue

Fluorescence lifetime maps for untreated ARBP and GE-ARBP showed homogeneous lifetime and intensity ratio maps in all three spectral bands. Due to the higher lifetime difference between cross-linked and untreated ARBP in SB3, this band was chosen as a basis for the selective sampling of regions of interest (ROIs) for Raman spectral acquisition. The Raman spectral features of untreated ARBP in Figure 3 confirmed the signature of type I collagen-rich tissue showing Raman bands at 814, 852, 935, 1001, 1242, 1272, 1453, and 1666 cm^−1^ at all the time points (Table 1). These bands were assigned to proline, hydroxy-proline, C-C_α_ stretch, amide III, C-H deformation, and amide I. The Raman spectrum of GE-cross-linked collagen has not been reported before likely due to the strong fluorescence emission from the GE cross-linker-derived pigment (generated by the partial absorption of the excitation laser light) that overwhelms the much weaker Raman signals. Figure 3 shows the changes occurring in the Raman spectra upon GE cross-linking at 0.5, 1, and 2.5 h of incubation.

At the 2.5 h time point, new Raman bands emerged at 1165, 1326, 1350, 1380, 1402, 1470, 1507, 1535, 1574, 1630, 1728, and 1741 cm^−1^ (Figure 3c), which may be related to blue pigmented GE cross-links. The spectral features of the GE cross-linker are resonance-enhanced and dominate the spectral features over type I collagen. The 2.5 h spectra had lower signal-to-noise ratio (SNR) due to the shot noise caused by the high fluorescence background. The Raman spectral changes observed at the 1 h time point were consistent with the Raman spectral changes observed at 2.5 h, except lower magnitude. The Raman spectra of GE-ARBP at 0.5 h resembled that of untreated ARBP with some GE cross-link-related bands that appear with a very weak SNR in comparison to the other two time points. The biological origin of the new bands was identified based on DFT simulations of the GE cross-linker molecule.

### 2.3. Density Functional Theory (DFT) Simulations Validate the Raman Signature of GE Cross-Linked Tissue

Density functional theory (DFT) is used in the field of vibrational spectroscopy to compute relaxed structures and vibrational modes of a molecule due to its comparatively low computational cost and good agreement with experimental references [36,37,38]. To date, GE cross-linked tissues have not been investigated using Raman spectroscopy. Hence, we used the DFT Raman simulation of the GE cross-linker to validate the experimental results obtained from GE-ARBP. The GE cross-linker is a dimerized genipin molecule that forms when GE reacts with the primary amino acids (Lys, H-Lys) of collagen (Figure 4a) [25,39]. The DFT-simulated GE dimer cross-linker showed prominent Raman bands at 1175, 1323, 1350, 1370, 1402, 1440, 1467, 1507, 1535, 1574, 1630, 1711, and 1728 cm^−1^ as summarized in Table 1 and shown in Figure 4b. The bands were assigned to the vibrational modes with relatively large Raman scattering cross section. The low wavenumbers 1175–1370 cm^−1^ are typically in-plane vibrations of the ring system, for example, the mode at 1350 cm^−1^ (Figure 4c). The modes at 1535 (Figure 4d) and 1574 cm^−1^ represent vibrations of the 5-membered ring system, in-plane motion of the atoms, and strong ring distortion, respectively. The intense band at 1630 cm^−1^ provides an in-plane motion of the 6-membered ring of the cross-linker (Figure 4e). The GE cross-linker also has C=O vibrations at 1711 (Figure 4f) and 1728 cm^−1^ and the splitting of the frequencies might be due to weak coupling between the monomers and slight differences in the chemical environment of C=O groups.

The Raman spectra (Figure 3b,c) and difference spectra (Figure 5c) of GE-ARBP at 1 and 2.5 h show strong Raman bands at 1326, 1350, 1402, 1507, 1535, 1574, 1630, and 1728 cm^−1^ that coincide with the DFT simulation results of the GE cross-linker. The bands at 1165 and 1380 cm^−1^ have an offset of 10 cm^−1^ relative to DFT spectral bands (1175 and 1370 cm^−1^) and the band at 1470 cm^−1^ has an offset of about 3 cm^−1^ relative to the DFT spectral band (1467 cm^−1^). The C=O bands occur at 1711 and 1728 cm^−1^ in the DFT spectrum, whereas in the GE-ARBP Raman spectra, the bands are observed at 1728 and 1741 cm^−1^.

### 2.4. Correlation between Raman Spectra and Fluorescence Parameters

To increase the biochemical specificity of FLIm, we explored the relationship between the FLIm parameters and the Raman spectra of GE-ARBP with a multivariate multiple regression (mvmr). Two independent fluorescence variables, namely fluorescence lifetime and intensity ratios, were predicted from the Raman spectra yielding two correlation coefficients (Figure 5a,b). Of note, based on the FLIm results, SB3 better captured the effects of the GE cross-linker, thus this analysis was performed with the FLIm parameters obtained in SB3. Both coefficients showed prominent Raman bands at 1165, 1326, 1350, 1380, 1402, 1470, 1507, 1535, 1574, 1630, 1728, and 1741 cm^−1^. The intensity ratio coefficient had positive Raman bands consistent with an increase in intensity ratio with increasing GE cross-linking (Figure 2d). Conversely, the lifetime coefficient had negative bands at equivalent frequencies consistent with lifetime decreases with increasing GE cross-linking (Figure 1f). The Raman difference spectra between GE-ARBP at different time points and their corresponding ARBP untreated control are shown in Figure 5c. These mimic the coefficients of the model having equivalent bands to those arising due to GE cross-linking, which serves as evidence that the changes in the fluorescence lifetime and intensity ratio of GE-ARBP observed in SB3 are mainly due to the GE cross-linker. The mvmr model could predict lifetimes and intensity ratios of test data with an R^2^ = 0.94 (Figure 5d) and R^2^ = 0.92 (Figure 5e), respectively.

## 3. Discussion

FLIm and Raman spectroscopy provide complementary biochemical information in a non-destructive and label-free manner. FLIm is a fast imaging technique that has modest biochemical specificity. Combining it with Raman spectroscopy increases the biochemical specificity of the overall technique. When GE reacts with collagen, it forms intra- and intermolecular cyclic cross-links with a blue pigmentation that stains the extracellular matrix (ECM). The results of the current study demonstrate that combined fiber-based FLIm and Raman spectroscopy could detect the effect of the cross-linker on the tissue specifically and identify the biochemical origin of the GE cross-linker.

Current results demonstrate that fluorescence lifetime is sensitive to the formation of GE cross-links. Specifically, the fluorescence lifetime steadily decreased with increased cross-linking (e.g., 5.3 ns in the untreated tissue versus 2.1 ns after 2.5 h cross-linking in SB1). This trend was observed for all SBs. Upon 355 nm excitation, the fluorescence emission spectrum of ARBP and GE-ABRP exhibited an emission maximum at 390 and 440 nm, respectively. The redshift has been associated to the GE cross-linker that generates a blue pigment as a by-product. The fluorescence intensity ratio measured with FLIm reflected this redshift in GE-ARBP (SB3). Such spectral change was not observed when ARBP was cross-linked with glutaraldehyde because no pigments resulted upon the cross-link formation [18].

Raman spectral measurements provided insight on the GE cross-linking and the associated pigment. To the best of our knowledge, there was no Raman spectral signature of genipin on GE-ARBP reported in the literature, likely due to the dominance of the blue pigment fluorescence. Therefore, we validated the experimental Raman spectra of GE-ARBP with DFT simulations of the GE cross-linker. The spectral features of the GE cross-linker were resonance-enhanced and dominated the spectral features over type I collagen. The effects of the blue pigment on the Raman spectra were seen on the difference Raman spectra of GE-ARBP, especially for the higher cross-linking time points (2.5 and 1 h) (Figure 5c). This characteristic blue pigmentation induced by GE cross-linking is possibly formed through a series of oxygen radical-involved polymerization and dehydrogenation of several intermediary pigments [25,31,40,41].

A multivariate multiple regression (mvmr) model was established to identify the relationship between FLIm parameters and Raman spectra. This step was used to enhance the biochemical specificity of the fluorescence parameters by associating them with biochemical changes in the tissue upon cross-linking, provided by the Raman spectra. The lifetime and intensity ratio coefficients obtained from the mvmr model explain the lifetime- and intensity ratio-related changes in SB3 (Figure 5a,b). The decrease in fluorescence lifetime reflects the biochemical changes induced by the GE cross-linker and the increase in intensity ratio is correlated to the appearance of the pigment.

One of the limiting factors with Raman spectroscopy was correcting the high fluorescence background, especially of GE-ARBP at 2.5 h. The sensitive nonlinear iterative peak (SNIP) baseline correction that we used could lead to overfitting or underfitting of the Raman spectra. In future applications, such a high fluorescence background could be tackled by using shifted excitation Raman difference spectroscopy (SERDS), a technique that can extract spectral information from highly fluorescent backgrounds [42,43,44].

In this study, we demonstrated that rapid imaging with FLIm and localized exploration with Raman spectroscopy can be used to monitor biochemical changes occurring during GE cross-linking of collagenous tissue. Furthermore, for rapid assessment, FLIm may be sufficient to track the evolution of specific metrics after a relationship is established between the FLIm parameters and the Raman spectra. The precise molecular signatures of Raman spectroscopy assign complementary meaning to the FLIm changes for an overall approach with enhanced biochemical specificity. This methodology proved useful for monitoring bioengineering processes of tissue modifications needed for biomedical applications [18], for example, to generate tissue grafts for regenerative medicine. Cardiovascular grafts are commonly cross-linked with glutaraldehyde to ensure biostability and compatibility with the host [45,46]. However, this fixative is cytotoxic and leads to eventual graft failure due to calcification [23]. Alternative natural and less cytotoxic fixatives like GE deserve further study. The ability of the FLIm-Raman approach to monitor the effect of GE on tissue non-destructively will ultimately aid the development processes of grafts with improved clinical outcomes.

## 4. Materials and Methods

### 4.1. Pre-Processing of the Bovine Pericardium (BP)

Bovine pericardium (BP) (Spear Products, Coopersburg, PA, USA) was antigen-removed and decellularized [47] and frozen at −80 °C until use. For the genipin cross-linking experiment, strips of antigen-removed bovine pericardium (ARBP) were thawed and further cut into 12 pieces of about 1 × 1 cm. The experimental procedure is depicted in Figure 6a.

#### Genipin (GE) Cross-Linking of ARBP

The 12 ARBP pieces were divided into four groups based on GE cross-linking incubation times of untreated, 0.5, 1, and 2.5 h. The treatment samples (N = 3 per group) were incubated in 0.25% GE in phosphate-buffered serum (PBS) at 37 °C, and the match controls (N = 1 per group) were incubated in PBS at 37 °C. Upon cross-linking, all samples were thoroughly rinsed in PBS for 3 h. Photographs of untreated ARBP and GE-ARBP at 0.5, 1, and 2.5 h are shown in Figure 6b.

### 4.2. FLIm—Raman Multimodal Fiber Probe

The fluorescence and Raman spectral acquisition was achieved by using a combined FLIm-Raman probe detailed on US patents 8.175.423 and 8.702.32123. Briefly, the optical fibers in the probe were made of UV low-OH fluorine-doped fused silica with 300 µm diameter. In the fiber bundle, the central fiber was dedicated to FLIm excitation (355 nm) and the collection of tissue autofluorescence, with no additional filters at the distal end. One of the peripheral fibers delivered the Raman excitation laser (785 nm) and had a low-pass filter at the distal end to reduce background. The Raman collection was mediated through seven peripheral fibers, covered with a U-shaped high-pass filter that rejected the laser line. All the fibers were tightly packed inside a stainless steel 14-gauge extra thin wall needle tube (1.82 mm ID, 2.1 mm OD). A 0.2 NA lithium-doped grin lens was mounted at the distal end of the fiber probe as an open tube attachment glued to the fiber bundle that provided non-contact measurements with a working distance of 1 to 1.5 mm. A schematic of the instrument can be found in [18].

### 4.3. FLIm—Raman Spectroscopy Instrument Design

#### 4.3.1. FLIm System

The FLIm system used in this study has been described previously [48]. Briefly, a frequency tripled Nd: YAG 355 nm laser (STV-02E-1 × 0, TEEM Photonics, Grenoble, France; 1.2 µJ pulse energy, 4 kHz repetition rate) was used for FLIm excitation. The autofluorescence signal was guided to a wavelength selection module comprised of a set of dichroic mirrors and bandpass filters that separate the tissue autofluorescence in four spectral bands (SB), namely SB1 = 380–400 nm, SB2 = 415–455 nm, SB3 = 465–553 nm, and SB4 = 572–642 nm. Only SB1, SB2, and SB3 were used in this study as they cover type I collagen and genipin cross-linked collagen emission upon 355 nm excitation. The fluorescence signal from each spectral band was temporally separated using optical fiber delay lines of 1, 13, 25, and 37 m for each band, respectively. A single microchannel plate photo-multiplier tube (MCP-PMT, R3809U-50, Hamamatsu, Japan) detected the fluorescence decays that were further sampled at 12.5 GS/s by a high-speed digitizer (3 GHz bandwidth; PXIe—5185, National Instruments, Austin, TX, USA). Using the FLIm system, images of size 22 × 20 mm were acquired within 5 min.

#### 4.3.2. Raman Spectrometer

The seven Raman collection fibers were connected to a Raman spectrometer (LS-785, Princeton Instruments, Acton, MA, USA) equipped with an 830 grooves/mm reflective grating and a front-illuminated, open electrode CCD camera (PIXIS -256B, Princeton Instruments, Acton, MA, USA). The CCD was thermoelectrically cooled to -70 °C and the signal was collected with full-range vertical binning. The Raman laser consisted of a 785 nm multimode laser (0811A100-B model/Ocean optics, Innovative Photonics solutions, NJ, USA) with excitation power fixed at 93 mW. Raman spectra were recorded with 3 s exposure time and 10 accumulations per pixel. The spectral range spanned from 200 to 3500 cm^−1^, with a spectral resolution of 15 cm^−1^.

#### 4.3.3. Fluorescence Spectrometer

The fluorescence spectra of 2.5 h cross-linked GE-ARBP and untreated ARBP was measured using a spectrophotometer (SpectraMax M5, Molecular Devices, San Jose, CA, USA) equipped with a PMT detector. A 355 nm excitation wavelength was used to excite the tissue and the emission spectrum was recorded between 380 and 650 nm, with a step size of 10 nm. The spectra were normalized to the maximum intensity of the emission spectrum.

### 4.4. Data Analysis

#### 4.4.1. FLIm Data Analysis

Both spectrally resolved time-decay parameters and spectral ratios derived from the three spectral bands of the wavelength selection module were analyzed. To evaluate the fluorescence decay characteristics, the FLIm data were processed based on a constraint least-square deconvolution with Laguerre basis functions as described previously [49]. Intensity ratios were calculated by dividing the fluorescence intensity of each spectral band by the sum of fluorescence intensities in all three spectral bands. Data are presented as the mean and standard deviation from pixels within the selected ROI of 1.5 × 1.5 mm from cross-linked samples (GE-ARBP) per each timepoint and untreated ARBP samples.

#### 4.4.2. Raman Data Analysis

All the Raman spectra were pre-processed using hyperSpec and MALDIquant packages in R. Spectra were truncated at the lower and higher wavenumber regions for final Raman spectra spanning from 800 to 1760 cm^−1^. All the spectra were intensity and wavelength calibrated. The spectra were background corrected using the SNIP algorithm [50] that used the SNIP function from the MALDIquant package in R [51]. All the Raman spectra were smoothed with the spc.loess (wavelength interpolation) function in hyperSpec [52]. The Raman spectra were normalized to the high SNR band at 1453 cm^−1^. This band was assigned to the CH_2_ vibrations of collagen and elastin, which remained invariant during the GE cross-linking process.

The Raman difference spectra for each time point (0.5, 1, and 2.5 h) were calculated by subtracting the GE-ARBP spectrum with the untreated ARBP spectrum. The mean and standard deviation were calculated from all collected spectra.

#### 4.4.3. Multivariate Multiple Regression

The regression model used for the FLIm-Raman comparison can be expressed in the matrix form
(1)Y=Xβ+ε
where *Y* contained two columns with the SB3 lifetimes and SB3 intensity ratios; *X* was the design matrix and consisted of 216 × 264 Raman intensities and wavenumbers; and *β* was a 2 × 264 matrix with the coefficients of SB3 lifetimes and intensity ratios. The number of components for the model was optimized by calculating the root mean squared error of the predictions. The model was trained with 66.6% of the experimental data and a “leave one out” cross-validation scheme was used for evaluation. The established model was tested with the remaining 33.3% of the data.

### 4.5. DFT Calculations

Density functional theory (DFT) was used to compute relaxed structures and vibrational modes of the GE cross-linker molecule. The B3LYP functional and 6-31G* basis set were employed, as implemented in the Q-Chem 5.1.2 software package [53]. This theory is commonly employed in the context of simulation of Raman signals since it nicely balances computational cost and accuracy [37,38]. The system was modeled as a positively charged singlet cation. Linkage to bovine pericardium collagen strands was taken into account by attaching methyl groups at the respective positions. The environment was incorporated via a polarizable continuum model (PCM) with a dielectric constant of 78.39 (water at 25 °C). Vibrational frequencies were scaled by a factor of 0.97 to account for systematic errors of the employed methodology [54].

## 5. Conclusions

Our results demonstrate that FLIm and Raman spectroscopy can detect the biochemical changes in the tissue induced by GE cross-linking. The resonance Raman spectra of the GE cross-linker were validated by DFT simulations, which is an appropriate approach to determine the unknown spectral origin of a molecule. Multivariate multiple regression established a relationship between the optical parameters of FLIm and Raman spectroscopy enhancing the biochemical specificity of the FLIm parameters to characterize the tissue. The main changes that occur in GE-ARBP were related to the GE cross-linker and its induced pigmentation. Additional experiments are due to further characterize the structural changes occurring in collagen molecules upon GE cross-linking. This multimodal imaging technique is viable to probe biochemical and structural changes during the tissue bioengineering process. Additionally, due to its ease of implementation with compact optical fibers, this approach is compatible with flat and intraluminal imaging that can be performed in vitro and in vivo. GE is a natural cross-linker with superior non-cytotoxic properties than standard fixatives such as glutaraldehyde. The ability to monitor the degree of cross-linking will help produce engineered tissue alternatives with improved clinical outcomes in regenerative medicine applications.

## Figures and Tables

**Figure 1 molecules-25-03857-f001:**
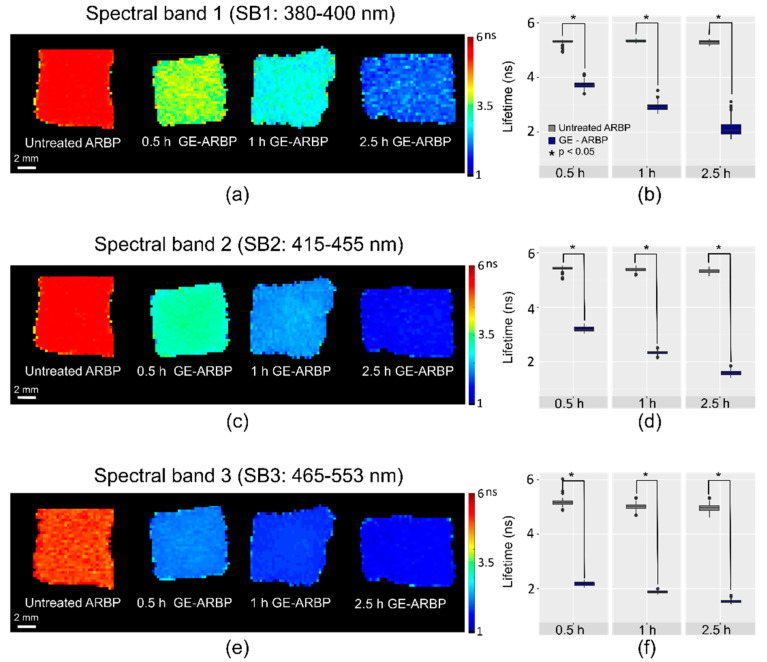
Fluorescence lifetime maps for GE-ARBP (0.5 h, 1 h, 2.5 h), and corresponding untreated ARBP in spectral band (SB) 1 (**a**), SB2 (**c**), and SB3 (**e**). Boxplots with the fluorescence lifetime mean and standard deviation of the control ARBP and treated GE-ARBP tissue pieces using three replicates for every time point in SB1 (**b**), SB2 (**d**), and SB3 (**f**). A paired sample Mann–Whitney U-test in each spectral band indicated a statistical difference between treated and control groups at each time point (* *p* < 0.05).

**Figure 2 molecules-25-03857-f002:**
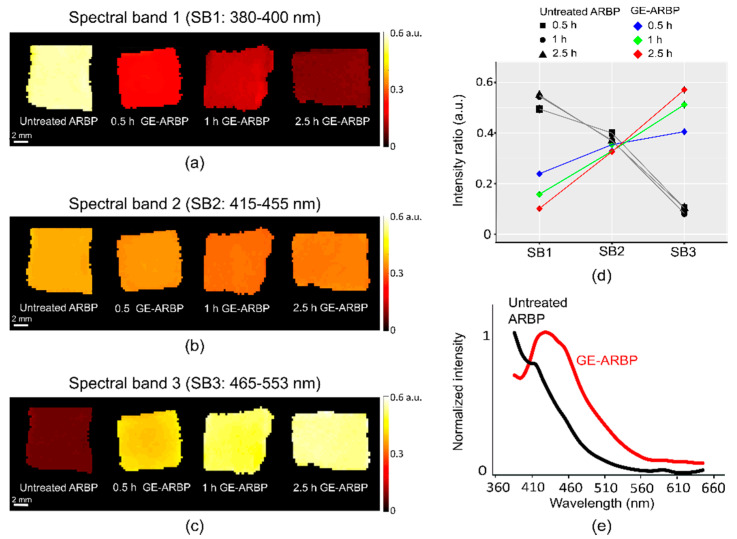
Intensity ratio maps of GE-ARBP (0.5 h, 1 h, 2.5 h) and corresponding untreated ARBP in SB1 (**a**), SB2 (**b**), and SB3 (**c**). (**d**) The quantified mean and standard deviation of fluorescence intensity ratios of GE-ARBP and corresponding untreated ARBP in SB1, SB2, and SB3. (**e**) Fluorescence emission spectra of untreated ARBP and GE-ARBP (2.5 h) upon 355 nm excitation.

**Figure 3 molecules-25-03857-f003:**
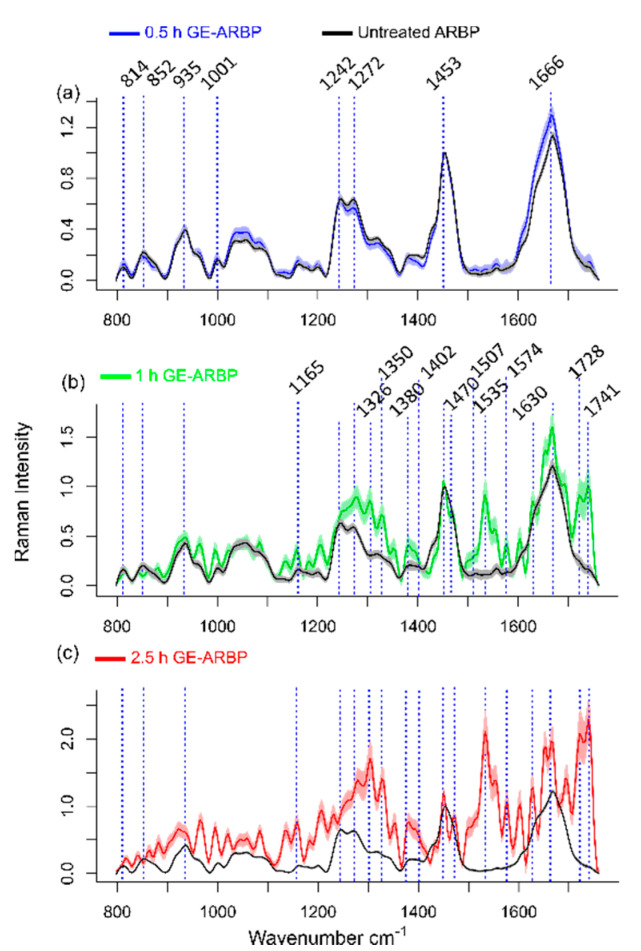
Raman spectra of untreated ARBP and GE cross-linked ARBP at 0.5 (**a**), 1 (**b**), and 2.5 h (**c**). Solid lines represent the mean and shaded areas represent the standard deviation of the Raman spectra (GE cross-linked N = 3, untreated ARBP N = 1). The prominent Raman bands are highlighted.

**Figure 4 molecules-25-03857-f004:**
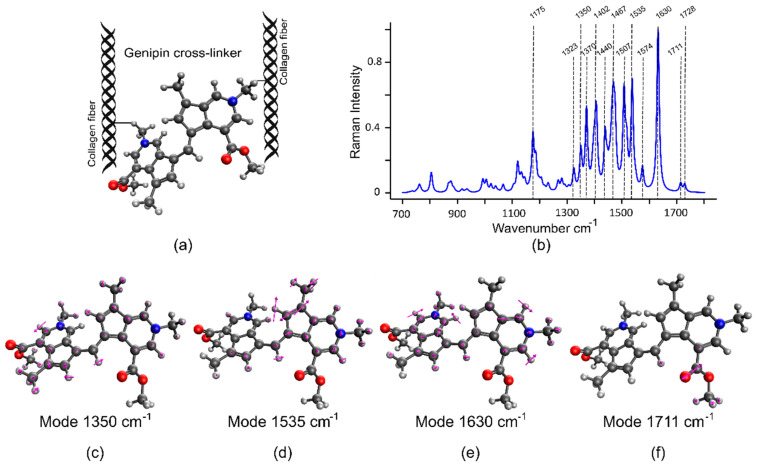
(**a**) Genipin cross-linker. (**b**) DFT-simulated Raman spectra of genipin cross-linker. Prominent Raman vibrational modes of genipin cross-linker: (**c**) mode 1350 cm^−1^, (**d**) mode 1535 cm^−1^, (**e**) mode 1630 cm^−1^, and (**f**) mode 1728 cm^−1^. Atom notation—blue: nitrogen, grey: carbon, red: oxygen, light grey: hydrogen.

**Figure 5 molecules-25-03857-f005:**
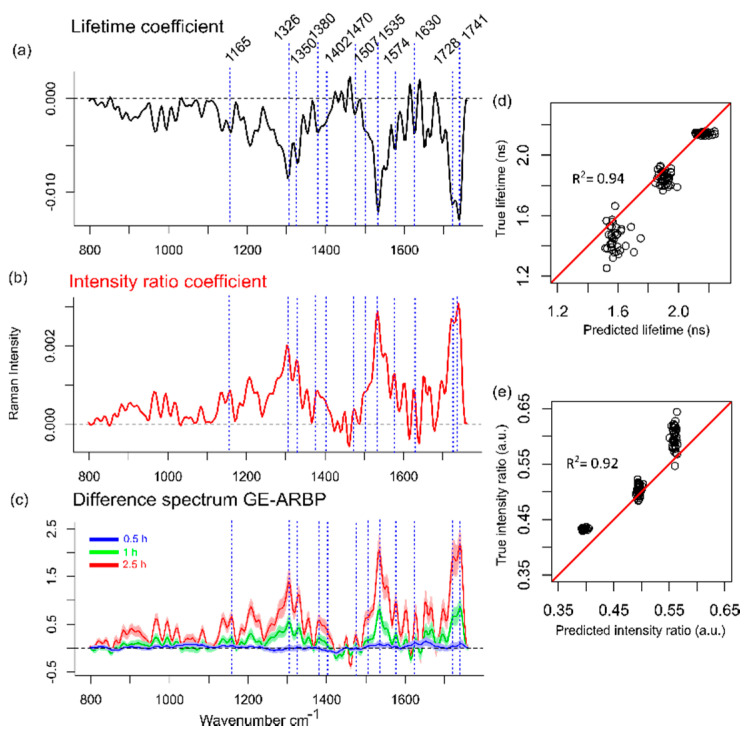
Multivariate multiple regression results show the correlation coefficients of fluorescence parameters lifetime (**a**) and intensity ratio (**b**) in SB3. (**c**) Mean (solid line) and standard deviation (shaded area) of the difference spectra between GE-ARBP and untreated ARBP at 0.5, 1, and 2.5 h. Experimental (or true) lifetime (**d**) and intensity ratio (**e**) values versus their predictions (red: corresponds to the ideal regression line with R^2^ = 1).

**Figure 6 molecules-25-03857-f006:**
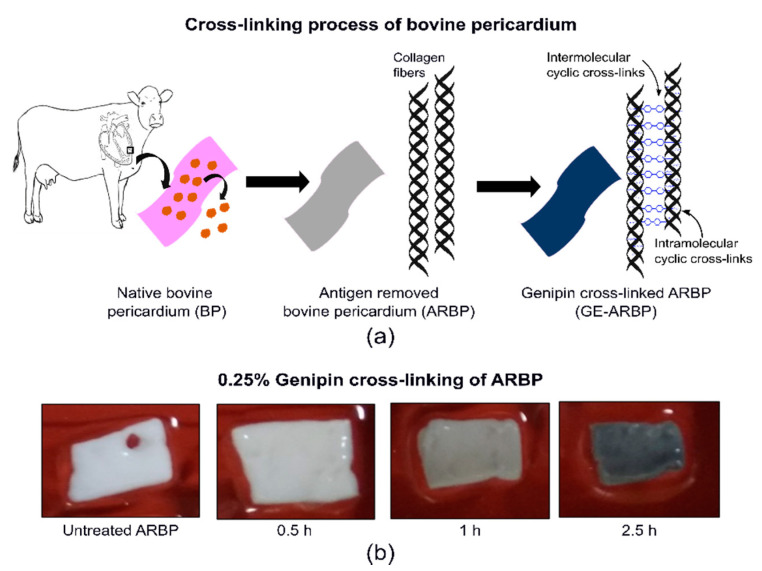
(**a**) Schematic of the pre-processing of the bovine pericardium and subsequent cross-linking with genipin. (**b**) Photographs of genipin cross-linked bovine pericardium at different time points.

**Table 1 molecules-25-03857-t001:** Band assignments for the Raman spectra detected in **GE-ARBP (in bold)** and untreated ARBP.

Wavenumber (cm^−1^)	Band Assignment	Reference
814	Proline	[18,35]
852	Hydroxy-proline	
935	C-C α stretch	
1001	Phenylalanine	
1242	Amide III	
1272		
1165	**In-plane vibrations of the ring system of GE-cross-links**	**DFT**
**1175**		
**1323**		
**1350**		
**1370**		
1380		
1400	**In-plane ring system vibration and bending of motion of CH_3_ group**	**DFT**
**1402**		
**1440**		
**1467**		
1470		
1453	CH_2_ collagen and elastin	[35]
**1507**	**In-plane motion of atoms in the 5-membered ring system**	**DFT**
**1535**		
**1574**	**Strong ring distortion of the 5-membered ring system.**	
1630	**Strong in-plane motion of the 6-membered ring**	
1666	Amide I	[18,35]
**1711**	**C=O**	**DFT**
**1728**		
1741		

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
