# Peer review of "FLIm and Raman Spectroscopy for Investigating Biochemical Changes of Bovine Pericardium upon Genipin Cross-Linking"

_molecules, 2020, doi:10.3390/molecules25173857_

Round 1

Reviewer 1 Report

This paper describes the monitoring of the degree of GE cross-linking of decellularized and antigen-removed bovine pericardium (ARBP) using non-destructive approaches based on fluorescence lifetime imaging and Raman spectroscopy, which detects the biochemical changes that occur in the tissue during the cross-linking process.

The paper is well researched, methodology is novel and well described and the results are convincing. I think it will advance the field, so I would recommend publishing it with minor revisions:

I would suggest carefully rewriting the abstract. There are some missing/oddly placed commas and some sentences can be rephrased to better capture the essence of the paper, in my opinion. Perhaps, the information contained in the sentence in the beginning of my review can be explicitly stated in the abstract…

Reviewer 2 Report

In the manuscript entitled “FLIm and Raman spectroscopy for investigating biochemical

changes of bovine pericardium upon genipin cross-linking”, by T. Shaik et al, are reported the optical property changes of an antigen-removed bovine pericardium tissue sample due to the interaction with Genepin, a natural cross linker. The biochemical modifications were, in particular, analyzed by using a system that permit to perform on the same sample fluorescence lifetime imaging and Raman spectroscopy.

Despite the particularity of the sample under investigation, the results reported in the manuscript are interesting for the non-destructive monitoring of tissues in bioengineering field.

The introduction well reports on the need for tissue engineering applications of non-destructive investigation techniques, like FLIm, together with biochemical specificity of Raman spectroscopy.

The experimental methodology and results presentation are adequate and well presented.

The spectroscopic results from Raman experiments were, moreover, supported by density functional theory (DFT) simulations and correlated to fluorescence lifetime results with a multivariate multiple regression approach.

For these reasons, in my opinion, the manuscript is suitable for publication on Molecules.
